# Unveiling the e-Servicescape of ChatGPT: Exploring User Psychology and Engagement in AI-Powered Chatbot Experiences

**DOI:** 10.3390/bs14070558

**Published:** 2024-07-02

**Authors:** Minseong Kim

**Affiliations:** Department of Management & Marketing, College of Business, Louisiana State University Shreveport, Shreveport, LA 71115, USA; minseong.kim@lsus.edu

**Keywords:** e-servicescapes, ChatGPT, chatbot, environmental psychology theory, artificial intelligence

## Abstract

This study explores the psychological motivations that drive ChatGPT users to embrace and sustain the use of such technology based on the fundamental notion of the environmental psychology theory, including servicescapes. To do so, this study delves into the influence of ChatGPT’s e-servicescapes on users’ emotional states and intention to engage with ChatGPT for decision-making processes. This study conducted an online survey among ChatGPT users in the United States. Structural equation modeling revealed that negative emotions were significantly influenced by various e-servicescape sub-dimensions, including security, visual appeal, entertainment value, originality of design, and social factors. Positive emotions, on the other hand, were influenced by factors such as visual appeal, customization, interactivity, and relevance of information. Both positive and negative emotions significantly affected user satisfaction, which, in turn, shaped their behavioral intention to engage with ChatGPT. This study contributes to the understanding of digital environmental psychology and chatbots by extending the notion of e-servicescapes to the context of AI-based services. It underscores the significance of e-servicescapes in shaping user experiences and provides valuable insights for business scholars and marketing practitioners.

## 1. Introduction

Artificial intelligence (AI) technology has become increasingly pivotal in providing users in physical and digital environments with unique and personalized experiences, as evidenced in empirical studies by Baek et al. [1], Baek and Kim [2], and Hu and Lu [3]. As elucidated by Ashfaq et al. [4], the deliberate deployment of generative AI chatbots, leveraging machine learning and natural language processing, has engendered a profound transformation. This transformation extends not only to how companies engage with their prospective and existing clientele but also to the decision-making processes of consumers themselves [2,3,4]. OpenAI’s ChatGPT, an advanced AI-powered chatbot, made its debut in November 2022 and has since attracted considerable attention from a broad spectrum of users, scholars in the field of business, and marketing practitioners. Its acclaim rests upon its ability to generate text responses, closely emulating human language and furnishing users with rapid access to essential information [2,5,6].

Diverging from traditional chatbots limited to fixed responses, ChatGPT exhibits a dynamic repertoire, capable of swiftly addressing inquiries, summarizing documents, composing essays, and generating comprehensive content, spanning social media narratives, advertising copy, and programming code tailored to specific languages [5,7]. This versatility is underpinned by the continuous refinement of ChatGPT’s capabilities through rigorous data-driven training, endowing it with conversational adeptness and a broad knowledge base [8]. Remarkably, ChatGPT also possesses a creative dimension, occasionally proffering responses that convincingly mimic human discourse and exude a sense of authority [2,7,8]. Nevertheless, a notable research lacuna exists concerning the empirical drivers motivating users to embrace and sustain the utilization of generative AI chatbots like ChatGPT. This knowledge gap persists despite the widespread adoption and popularity of such technology [6,7]. The investigation of these hitherto uncharted psychological motivations assumes paramount significance for both business researchers and marketing practitioners. It furnishes invaluable insights into user behavior and offers strategic guidance for the continued enhancement of the development and deployment of generative AI chatbots, exemplified by ChatGPT [2,6].

Prior studies in the domain of chatbots have predominantly focused on identifying the motivational factors that impact users’ continued utilization of such technology in their decision-making processes [2,7,9,10]. However, these prior investigations have primarily revolved around the technological functionalities of chatbots, neglecting the psychological dimensions experienced by users [9,10,11]. In contrast, our research departs from prior conceptual and empirical inquiries by closely examining the impact of ChatGPT’s virtual structure and design on users’ psychological responses, emotional states, and their sustained intention to engage with this technology for their daily decision-making processes [5,11]. More precisely, the transition from conventional search engine systems, such as Google, to ChatGPT constitutes a transformation anchored in the concept of the “servicescape” that users encounter from a psychological perspective. In our research, we place particular emphasis on this facet of ChatGPT’s virtual service in addition to its technological aspects.

Servicescapes, defined as the amalgamation of tangible and physical cues, function as a representation of an organization to its clientele. Within the domains of marketing and consumer behavior, servicescapes are widely acknowledged as fundamental catalysts influencing customer satisfaction, behavioral intent, and even tangible behaviors. This foundational concept finds its roots in environmental psychology theory [12,13,14,15]. According to this theory, the tangible and physical cues present in both physical and virtual environments exert a discernible impact on users’ emotional states, behavioral intentions, and actual behaviors, encompassing tendencies such as avoidance or approach [12,14,16,17,18]. Expanding upon the underpinnings of environmental psychology theory, previous research has ventured into the realm of digital environments encountered by users, often referred to as the “e-servicescapes”. This exploration has sought to unravel the constructive influence of these digital environments on user perceptions, sentiments, intentions, and behaviors concerning brands and products [12,14,18]. Similarly, when users engage with ChatGPT as part of their decision-making process, the significance of the e-servicescape comes to the forefront. This concept assumes a pivotal role in shaping users’ emotional states and their behavioral intentions during their digital interactions with ChatGPT, adhering to the tenets of environmental psychology theory [12,16].

Therefore, building upon prior conceptual and empirical inquiries into e-servicescapes [12,15,17,18,19], our study seeks to extend and apply the foundational concept of servicescapes to the context of chatbots, notably ChatGPT. These endeavors aim to provide marketing scholars and practitioners with a more comprehensive and holistic understanding of chatbots. In pursuit of this academic objective, our research delves into how users’ perceptions of ChatGPT’s servicescapes impact their emotional states, encompassing both positive and negative dimensions, during their virtual interactions with this technology [12]. These emotional states subsequently exert an influence on user satisfaction with ChatGPT and their intent to sustain engagement with it. In this context, our research contributes to the extant body of understanding and knowledge on chatbots and digital environmental psychology, expanding its relevance to the domain of AI-based services. This expansion is achieved through the conceptualization and operationalization of a multi-dimensional framework for ChatGPT servicescapes.

## 2. Literature Review 

### 2.1. AI-Driven Chatbots and Human Behavior

AI chatbots have increasingly become integral to various domains, impacting human behavior in decision-making processes, privacy concerns, and emotional responses [2]. Understanding these impacts requires a thorough examination of existing literature on AI chatbots, particularly in the context of generative AI and social chatbots like Replika AI. Generative AI chatbots, including platforms like Replika AI, represent a significant advancement in AI applications. These chatbots are designed to engage users in meaningful and human-like conversations, often resulting in deep emotional interactions.

Drouin et al. [20] investigated the nature of user interactions with social chatbots like Replika AI. Their study revealed that these interactions often evoke strong emotional responses, with some users forming emotional bonds with the chatbots akin to those with human friends. This emotional engagement highlights the potential of social chatbots to provide companionship and emotional support, especially for individuals seeking interaction and connection. Wang et al. [21] further examined the psychological effects of prolonged engagement with AI chatbots. They found that while these interactions could lead to positive outcomes, such as improved mood and reduced feelings of loneliness, there are also potential negative effects, such as dependency and emotional over-reliance on the chatbot. This duality underscores the importance of designing AI chatbots that can support healthy emotional interactions without fostering dependency.

AI chatbots influence decision-making processes by providing users with quick access to information and personalized recommendations. Liu and Ma [9] examined how AI chatbots affect consumer decision-making by offering tailored suggestions based on user preferences and past behaviors. Their study found that users are more likely to make confident and informed decisions when interacting with AI chatbots that provide relevant and accurate information. The impact of AI chatbots on user behavior and engagement has been a focal point in recent research. Cheng and Jiang [22] investigated how AI-driven chatbots influence user loyalty and continued use by examining gratifications, perceived privacy risk, satisfaction, and loyalty. Their findings indicate that user engagement is significantly influenced by the chatbot’s ability to meet user needs and provide a satisfying interaction experience.

Understanding the link between emotional responses and behavioral intention is critical for designing effective AI chatbots. Ladhari, Souiden, and Dufour [23] highlighted the importance of positive emotional experiences in fostering user satisfaction and loyalty. They found that positive emotions such as joy and excitement significantly enhance users’ intentions to continue using AI chatbots. Conversely, negative emotional responses such as frustration and annoyance can deter users from engaging with AI chatbots. For example, Kim [12] examined the effects of negative emotions on user satisfaction and found that these emotions significantly reduce user engagement and increase the likelihood of discontinuing use. Their research underscores the importance of addressing sources of negative emotions, such as poor usability and irrelevant information, to improve user satisfaction and retention.

### 2.2. A Theoretical Background: The Theory of Environmental Psychology

Environmental psychology theory, as applied to digital environments, provides a foundational framework for understanding how virtual structures influence user emotions and behaviors. In accordance with the foundational principles of environmental psychology expounded by Kim [12], Tankovic and Benazic [19], and Tran and Strutton [18], it is established that human behaviors are markedly influenced by the environments in which individuals interact and engage. To elucidate further, it is pertinent to underscore that all tangible and physical stimuli within an environment assume a pivotal role in molding individuals’ emotional states and exerting a comprehensive influence on their decision-making processes [15,17]. In more simplified terms, employing holistic approaches, individuals are predisposed to evaluate various facets of their surroundings, encompassing an assessment of the emotional and subjective dimensions (grounded in their prevailing emotional state within the environment), psycho-physical dimensions (informed by their responses to environmental stimuli), cognitive dimensions (influenced by the information presented about the environment), and experiential dimensions (shaped by their interactions within the environment) [12,24]. Consequently, the central tenet of this theory postulates that individuals actively seek out and appraise environmental cues with consideration for their tangible attributes and emotional states, thereby underscoring the prominence of holistic perspectives [19].

Within the ambit of commerce, consumers are routinely exposed to a profusion of visually presented information pertaining to products, services, brands, and corporate entities [12]. To contend with the deluge of information, consumers employ the construction of a cognitive map, which assumes a pivotal role in their decision-making processes [25]. Notably, the construction and utilization of this cognitive map by consumers are intricately shaped by both the external and internal facets of their environments, encompassing what is commonly referred to as “servicescapes” [13,25]. The experiential and visual facets of servicescapes, whether within a physical or virtual business milieu (e.g., brick-and-mortar retailers and online websites), can exert a profound influence on consumer choices, satisfaction with specific products or services, as well as their emotional and perceptual assessments of service quality [26]. As such, the concept of an “e-servicescape” has emerged to denote the virtual environments that users encounter, observe, and experience [12]. Analogous to their physical counterparts, websites or webpages, inclusive of their extrinsic and intrinsic environmental cues, wield a substantial impact on the emotions of their users and their formulation and utilization of cognitive maps in the decision-making process [25,26]. A comprehensive exploration of the notion of e-servicescapes shall be undertaken in subsequent sections.

### 2.3. e-Servicescape

The concept of “servicescape” was originally introduced by Baker [27] as a framework for characterizing the physical environments associated with services, which consumers assess based on factors such as design, ambiance, and social elements. Bitner [28] expanded upon this concept by introducing the term “servicescapes”, encompassing a broader spectrum of elements, including symbols, artifacts, signs, as well as ambient conditions, functionality, and layout. Extending this foundational idea, “e-servicescape” is defined as environment-oriented factors existing in the virtual realm when digitalized services are provided to users [29]. In the context of consumer behavior, Dassanayake and Senevirathne [30] define e-servicescape as the atmospheric environmental features of a virtual space with which consumers engage, interact, and experience. Similar to traditional brick-and-mortar retail establishments, websites and webpages are thoughtfully designed to evoke positive responses and foster favorable interactions with consumers [12,15,19].

Within the existing body of literature on online consumer behavior, the construct of e-servicescapes comprises a multitude of factors that focus on the virtual environment within a website or digital setting [26]. These factors can be categorized into two main components: traditional elements that are relevant in both traditional and digital store contexts and novel factors specific to the digital landscape. In line with the foundational servicescape concept, traditional retailers place emphasis on aesthetically pleasing factors such as design originality, visual appeal, and entertainment value [12]. Additionally, they consider aspects like customization, usability, functionality, interactivity, and the relevance of information [18,19]. The distinctive facets of e-servicescapes in the online environment revolve around security in financial transactions, including perceptions of ease of payment and security [12,18]. Furthermore, e-servicescape encompasses specific online attributes, such as user-friendly interfaces, speed, timeliness of information, privacy, search pathways, navigation, music, and customization [15,18]. Consumers tend to assess websites and online environments holistically, considering factors such as layout, security, functionality, visual appeal, and entertainment value concurrently [12,30].

However, previous empirical studies have not extensively delved into the conceptual dimensionality of e-servicescape within the context of chatbots, including aspects like symbolism and functionality within hotel mobile applications [31], the dimensionality of mobile app servicescapes in live streaming platforms [32], or the subdimensionality of e-servicescapes in fitness mobile applications [12]. While ChatGPT shares some digital environment-oriented features with service providers like webpages or websites, users’ preferences for virtual environments may diverge when interacting with AI-based chatbots. Consequently, our research aims to explore the sub-dimensionality of e-servicescapes, taking into account the unique characteristics of ChatGPT. This endeavor is intended to provide scholars in the chatbot domain with a comprehensive understanding of each sub-dimension of e-servicescapes.

### 2.4. Emotions

Emotions encapsulate a wide spectrum of both negative and positive affective states within individuals in a general context [12]. These emotional states are often linked to specific referents, such as objects, events, or individuals [33]. Individuals experiencing positive emotions typically engage in simplified decision-making processes and exhibit a propensity to deliberately allocate less time to decision-processing compared to those immersed in negative emotional states [12,23,34]. Over the past two decades, extensive research has consistently underscored the pivotal role of positive and/or negative emotions across various dimensions of consumer perceptions, attitudes, and behaviors. These dimensions encompass the evaluation of service/product quality, selection of service/product providers, prediction of repeat purchase behaviors, and the cultivation of loyalty toward a service/product [12,33,35,36]. Dubé and Menon [37] asserted that “Consumption emotions are the affective response to one’s perceptions of the series of attributes that compose a product or service performance” (p. 288). This assertion underscores the essential role of emotional facets in comprehending consumers’ assessments of their consumption experiences.

In the past three decades, previous research has applied and validated various emotional states experienced by consumers throughout the entire consumption process. For instance, emotions have been identified as eight primary states, including joy, acceptance, anticipation, fear, anger, sadness, disgust, and surprise [38]. These emotional states are often organized into circular patterns, allowing for combinations and transitions between them [12,36]. Empirically, the measures developed by Chebat and Slusarczyk [35], adapted from the work of Plutchik [38], have substantiated the significant role of consumer emotions in managing service recovery and customer complaints within a retailing and consumer service context. Furthermore, prior empirical studies have consistently affirmed the distinct impacts of consumers’ negative and positive emotional states on their actual behaviors and behavioral intentions [12,36,39]. For example, the investigation conducted by Machleit and Eroglu [40] provided evidence that consumers’ negative and positive emotional states exert separate effects on their behaviors and attitudes, each contributing different variances. Consequently, our research endeavors to categorize ChatGPT users’ emotional states into positive and negative aspects separately.

### 2.5. Satisfaction and Behavioral Intention

The term “satisfaction” is defined as “the overall psychological state that emerges when a consumer’s emotional response to unmet expectations is amalgamated with their prior sentiments concerning the consumption experience” [41] (p. 27). It is widely acknowledged as a pivotal determinant that significantly influences consumers’ intentions to engage in repeat purchase behaviors and foster loyalty [12,42]. Previous scholarly work has traditionally classified customer satisfaction into two primary categories: “transaction-specific” and “general overall satisfaction” [12,43,44]. Transaction-specific satisfaction pertains to a customer’s appraisal of a product or service subsequent to its purchase, utilization, and experiential engagement, whereas overall satisfaction represents a holistic rating provided by the customer, reflecting their comprehensive experiences [43,44]. Consequently, a sequence of discrete product or service interactions with providers over a specified time frame can cumulatively shape consumers’ overall satisfaction [36,43]. General overall satisfaction typically wields a more substantial impact on consumers’ favorable attitudes and behaviors, including loyalty, across a diverse spectrum of service and product domains. This prominence arises from the fact that it is typically informed by a composite of consumers’ historical transaction-specific interactions with specific services and products [12,36]. Notably, the study conducted by Fournier and Mick [45] consistently demonstrated that exclusive reliance on transaction-specific satisfaction might impose constraints on the conceptual breadth of research. In the service industry context, general overall satisfaction often aligns with the overarching assessment of service quality as perceived by consumers [12,44,46]. In contrast to measures concentrated on individual transactions or episodes, comprehensive evaluations tend to wield a more potent influence on consumer behaviors, thereby benefiting companies through increased repeat purchases and positive word-of-mouth recommendations [42,46]. Given that satisfaction is an outcome stemming from consumers’ experiences that span various stages of the decision-making process [44,46], gaining insights into users’ satisfaction with ChatGPT assumes paramount importance for computer programming developers.

Promoting positive consumer behaviors has emerged as a paramount objective for organizations striving to establish and sustain a competitive foothold within the dynamic contours of the contemporary market landscape [12,22,46]. Consequently, consumers’ proclivity to exhibit favorable behaviors toward a product, service, brand, or company—signifying a robust commitment to unwavering support and long-term patronage, impervious to situational variables and external competitive enticements—is of profound significance [47,48]. Furthermore, the favorable behavioral intentions of both prospective and extant consumers can manifest in multifaceted forms, including endorsements, positive word-of-mouth, and a willingness to incur higher costs [22,48]. Consequently, for developers of chatbot technologies, the retention of existing users and the reinforcement of their favorable behavioral inclinations represent pivotal strategies for attaining a competitive edge, particularly within the e-servicescape. In this highly competitive arena, where alternatives abound at the mere click of a button, these strategies assume heightened importance [22,47].

### 2.6. Research Hypotheses

The conceptual nexus between e-servicescapes and the emotional responses of users, both positive and negative, entails an exploration of how the multi-dimensional facets of the virtual environment can engender either favorable or unfavorable affective reactions during engagements within the digital milieu [12]. Initially, an aesthetically pleasing virtual structure or platform within a web application has the potential to elicit positive emotions among users, including sensations of pleasure and enjoyment, primarily due to its visual allure encompassing an attractive layout, graphical elements, and overall design aesthetics. This visual aesthetic appeal serves to augment user experiences within both physical and digital realms, thereby yielding constructive emotional responses [17,24]. Conversely, a digital environment characterized by an unattractive or cluttered design may act as a catalyst for negative emotions among users, eliciting sensations of annoyance or frustration and consequently impeding their capacity to engage effectively with the virtual milieu [12,24].

Secondly, the presence of user-friendly interfaces and efficient navigation mechanisms plays a pivotal role in cultivating positive emotions among users. When users can effortlessly complete tasks and interactions, they are more likely to experience a heightened sense of efficiency and competence, thus giving rise to emotions such as joy and happiness [49]. In stark contrast, a digital environment characterized by poor navigation or usability issues can lead users to experience disorientation and psychological impediments, thereby triggering negative emotions, including irritation and frustration [12,17,49].

Thirdly, virtual environments characterized by a high degree of interactivity and engagement have the propensity to evoke positive emotions among users, encompassing sensations of enjoyment, curiosity, and excitement. Interactive elements, personalized content, and the integration of chatbots, among other features, contribute to this effect [34]. Conversely, digital environments lacking engaging functionalities or interactivity may induce feelings of boredom or apathy as they fail to capture user interest, thereby inciting negative emotional responses [17].

Fourthly, the provision of high-quality, reliable information within a digital environment can foster user credibility and trust, thereby eliciting positive emotions such as reassurance and confidence. Users tend to feel secure when they perceive information as trustworthy and accurate [12,50,51]. However, a lack of transparency or the dissemination of unreliable information may provoke negative emotions, including anxiety or suspicion, leading users to feel misled or uncertain and engendering negative emotional reactions [50,51].

Lastly, when a digital environment tailors experiences or content to user preferences, it has the potential to evoke positive emotions, fostering a sense of appreciation and value among users. Conversely, intrusive or inadequate personalization efforts may elicit negative emotions, such as perceptions of privacy invasion or annoyance, prompting negative reactions by users who perceive personalization as irrelevant or intrusive [12]. Rajaobelina et al. [48] emphasize the importance of addressing privacy concerns and technology anxiety to build trust and loyalty among users of AI chatbots. Their work highlights the critical role of emotional factors in user satisfaction and engagement with AI technologies. In light of the foregoing, the following hypothesis is posited:

**H1:** 
*e-Servicescape has a significant effect on ChatGPT users’ positive and negative emotions toward ChatGPT.*


In this study, users’ emotional responses to their interactions with ChatGPT are examined in terms of both positive and negative dimensions. Firstly, positive emotions play a pivotal role in shaping users’ evaluations of their experiences and interactions with ChatGPT. This positive affective state contributes to higher levels of user satisfaction, primarily through the channels of enjoyment, perceived efficiency, and perceived value [12,23,34]. Consequently, when users experience positive emotions during their interactions and engagements with ChatGPT, they tend to exhibit greater engagement and investment in these interactions and experiences. This heightened engagement, in turn, leads to a sense of fulfillment and accomplishment, which contributes significantly to their overall satisfaction with ChatGPT [23,52]. Furthermore, it is noteworthy that users’ positive emotions can create a reinforcing feedback loop, where individuals reporting higher levels of satisfaction with ChatGPT are more inclined to offer positive feedback and recommendations to others, thereby further bolstering their own satisfaction with the system [34,53]. Conversely, users’ experiences of negative emotions, encompassing sentiments such as disappointment, irritation, or frustration, can exert detrimental effects on their satisfaction with ChatGPT. Specifically, these negative emotional states can serve as triggers for cognitive dissonance—a psychological state characterized by discomfort arising from incongruence between one’s expectations and actual experiences [12,53]. Users are prone to experiencing cognitive dissonance if ChatGPT fails to align with their initial expectations, resulting in diminished levels of satisfaction. Furthermore, negative emotions have the potential to amplify sentiments of dissatisfaction, prompting users who feel frustrated or disappointed to adopt a more critical stance toward ChatGPT’s performance, consequently further eroding their overall satisfaction [52,53]. In light of the above delineations, this research posits the following hypothesis:

**H2:** 
*A user’s positive and negative emotions during digital interactions with ChatGPT have a significant effect on satisfaction with ChatGPT.*


From the perspective of approach motivation, users’ positive emotions bear considerable relevance to their motivations, predisposing them to embrace and sustain engagement with ChatGPT. These positive emotional states engender a proclivity to pursue and perpetuate ongoing positive interactions [12,34]. Furthermore, as positive emotions augment users’ perceptions of the inherent value associated with ChatGPT utilization, individuals are inclined to intensify their intent to persist in employing this technology as they correlate such positive emotions with the system’s accrued benefits and utility [34,52]. Conversely, adopting an avoidance motivation standpoint, users’ negative emotions evoke avoidance tendencies, thereby reducing the likelihood of contemplating the discontinuation of their engagement with ChatGPT [12,53]. In the realm of psychology, these individuals tend to conflate negative emotions with a dearth of utility or advantages stemming from continued utilization of ChatGPT [23,53]. Additionally, users experiencing negative emotions during their interactions with ChatGPT may construe such engagements as obstructive rather than facilitative, thereby prompting a decline in their inclination to persist in its usage [23]. Consequently, the negative emotions experienced by users can function as a form of negative reinforcement, compelling them to eschew future encounters characterized by negative emotional experiences, thereby culminating in a cessation of interactions with this technology [12]. In light of these premises, our research posits the following hypothesis:

**H3:** 
*A user’s positive and negative emotions during digital interactions with ChatGPT have a significant effect on behavioral intention to continuously use ChatGPT.*


The link between customer satisfaction and behavioral intention has been substantiated through both cognitive and affective perspectives [12,54]. Firstly, from a cognitive vantage point, individuals’ behavioral intentions are influenced by their expectations regarding the outcomes of direct or indirect interactions with a product or service, along with the perceived value attached to these outcomes [55]. In the specific case of ChatGPT, users may harbor distinct expectations concerning the caliber and utility of their interactions with the AI, encompassing attributes such as helpfulness, accuracy, and responsiveness [47,54,56]. When these attributes fail to align with user expectations, it can engender a state of cognitive dissonance—a psychologically discomforting condition [12]. Consequently, users are predisposed to modify their behavioral intentions as a means of reconciling this cognitive dissonance, either by discontinuing or persisting in their utilization of ChatGPT [22,56]. Secondly, from an emotional perspective, the satisfaction of users can yield loyalty, while dissatisfaction has the potential to attenuate it, as posited by affective psychology theory. Specifically, satisfaction, characterized as a positive emotional state, tends to augment individuals’ proclivity and motivation to engage in behaviors pertaining to a specific product or service [22,47]. This influence is predicated on the recognition that emotions wield a substantial impact on decision-making processes, including the formulation of judgments and the selection of courses of action [47,54]. Accordingly, customer satisfaction with ChatGPT has the capacity to instill a positive emotional bias, thereby fostering more favorable attitudes and inclinations toward future usage of ChatGPT. In light of the aforementioned principles, our research advances the following hypothesis:

**H4:** 
*A user’s satisfaction with ChatGPT has a significant effect on behavioral intention to continuously use ChatGPT.*


## 3. Methods

### 3.1. Data Collection

Through the utilization of an online survey methodology, this research harnessed the resources of Amazon’s Mechanical Turk (MTurk) and Qualtrics to procure data from their panel databases. The subjects of our analysis were users of ChatGPT residing within the United States. The selection of ChatGPT users within this geographical context was deliberate, as it sought to mitigate potential influences stemming from cultural diversity and varying levels of societal norms on the respondents’ input [2]. This approach was consistent with prior research in the field of marketing, where MTurk and Qualtrics have been employed to explore the dynamics of user interactions with ChatGPT and chatbots, owing to the advantages they offer [2,57,58]. Participants were compensated for their time and effort. Each participant received $1.00 for completing the survey, which is a standard rate for surveys of this length and complexity on MTurk. This compensation was intended to incentivize participation without introducing undue influence [2].

Within our online survey platform, participants were afforded the opportunity to engage directly with ChatGPT. Specifically, participants were engaged in text-based tasks to gather insights into their emotional responses and satisfaction levels because this study specifically examined the text-based interactions with ChatGPT. This deliberate focus on text-based interactions ensures a detailed understanding of this particular feature while acknowledging that other modalities, such as voice, image, or video, were not included in this study. While ChatGPT has evolved through various versions, including GPT-3, GPT-3.5, and GPT-4, our research focuses on user interactions with the text-based features of ChatGPT version GPT-3.5 (i.e., great for everyday tasks). It is important to note that this scope excludes other modalities such as voice, image, or video capabilities. 

All participants provided informed consent before participating in the study. The consent form outlined the purpose of this study, the nature of the tasks involved, the voluntary nature of participation, and the measures taken to ensure data privacy and confidentiality. This study was conducted in accordance with the Declaration of Helsinki and received ethical approval from the Institutional Review Board (IRB) of Louisiana State University Shreveport (LSUS 2022-00024). Prior to proceeding to the main questionnaire, they were tasked with responding to an open-ended query soliciting their impressions of their interactions with ChatGPT. This technical, strategic maneuver was aimed at invoking participants’ recollections of both past and ongoing experiences with ChatGPT, thus enriching the quality of their responses [59]. 

To ensure the reliability of the dataset, all responses to the open-ended question underwent meticulous scrutiny by two established scholars and three graduate students well-versed in this domain. Brief backgrounds and relevant research experiences of the scholars are outlined below: (1) The first scholar is a professor of marketing at a public university in the United States with over 20 years of experience in digital marketing and consumer behavior. He has published extensively on topics such as online consumer behavior, digital advertising, and e-commerce; (2) the second scholar is an associate professor at a private university in South Korea with a research focus on digital marketing and social media analytics. He has authored several high-impact articles on the effectiveness of digital marketing campaigns and consumer engagement on social media platforms. His expertise in digital environments provided valuable insights for this study, and (3) the third group of contributors comprises three graduate students with various experiences in digital marketing and online consumer behavior. They have conducted numerous studies on user interactions with digital interfaces and the psychological effects of digital marketing under the guidance of their academic advisors. Consequently, the research yielded a total of 548 valid samples, following the exclusion of 22 instances characterized by insufficient responses, such as brief phrases like “it’s cool” or “no idea”. 

For a comprehensive overview of the participants’ demographic profiles, Table 1 provides a visual representation of the dataset used in the subsequent analyses. The demographics of our sample closely align with the reported demographics of ChatGPT users. According to Namepepper, in 2024 [60], the majority of ChatGPT users in the United States are aged 18–34, with a significant portion having at least a bachelor’s degree. Our sample from MTurk reflects these characteristics, as detailed in Table 1, where 71.4% of our respondents were aged 18–29, and 59.7% had at least some college education.

### 3.2. Measures

Our research conducted an exhaustive literature review on the concept of e-servicescapes encompassing both its theoretical dimensions and empirical operationalization. In the realm of conceptualization, we first synthesized the diverse dimensions and measurement approaches pertaining to e-servicescapes, as evidenced in prior empirical studies. Specifically, we drew insights from the works of Kim [12], Tankovic and Benazic [19], Teng et al. [17], Tran and Strutton [18], Wu et al. [24], and Yadav and Mahara [15]. Subsequently, in a conceptual inquiry, our research solicited the expertise of three scholars specializing in digital marketing. These scholars were presented with the synthesized dimensions and measures and tasked with evaluating the extent to which each measure from previous research in the domain of digital consumer behavior adequately reflected the context of chatbots. Their evaluations were structured on a scale ranging from 1, indicating “not reflected at all”, to 3, signifying “well reflected”, thereby establishing the content validity of each measurement. Consequently, a meticulous curation process yielded a total of 50 measurements, distributed across nine sub-dimensions that pertain to e-servicescapes, as the final inventory of survey items. This curation process involved the exclusion of measurements that received a low rating of 1 point from three scholarly evaluators. Subsequently, in alignment with empirical research practices, our study engaged online panels sourced from Qualtrics and MTurk to execute preliminary assessments through a pilot test. Within this phase, seven items were omitted based primarily on the results of an exploratory factor analysis, and the wording of the items was refined in response to participant feedback and responses. As a result, a refined set of 43 items was ultimately selected to serve as survey items, capturing the nine sub-dimensions characterizing e-servicescapes within the context of ChatGPT. 

(1)*Usability* refers to the ease with which users can navigate and interact with the ChatGPT interface. High usability ensures that users can efficiently and effectively use ChatGPT, leading to a positive user experience and higher satisfaction;(2)*Security* refers to the perceived safety and protection of users’ data and interactions with ChatGPT. Ensuring robust security measures enhances users’ trust and willingness to engage with ChatGPT;(3)*Visual appeal* refers to the aesthetic attractiveness of the ChatGPT interface. A visually appealing interface enhances user satisfaction and engagement by providing a pleasant user experience;(4)*Customization* refers to the extent to which ChatGPT can be tailored to meet individual user preferences. High customization allows users to personalize their experience, resulting in increased satisfaction and engagement;(5)*Entertainment value* refers to the extent to which ChatGPT provides enjoyment and amusement to users. High entertainment value can enhance user engagement and satisfaction by making interactions with ChatGPT enjoyable;(6)*Interactivity* refers to the degree to which ChatGPT allows users to interact and engage with its features dynamically. High interactivity enhances user engagement by allowing for more personalized and responsive interactions;(7)*Originality of design* refers to the uniqueness and creativity of the ChatGPT interface. A unique and original design can enhance user interest and satisfaction by providing a novel experience;(8)*Relevance of information* refers to the extent to which the information provided by ChatGPT is pertinent and useful to the user. High relevance of information ensures that users receive accurate and valuable content, resulting in increased satisfaction and trust;(9)*Social factors* refer to the extent to which ChatGPT provides a sense of human-like interaction and social presence. High social factors can enhance user engagement by making interactions with ChatGPT feel more personal and human-like.

Furthermore, our study encompassed the evaluation of outcomes associated with e-servicescapes. Specifically, we measured positive emotion and negative emotion, employing four items each, drawing from the works of Kim [12] and Kim and Stepchenkova [61], respectively. Additionally, we gauged satisfaction using three items sourced from the research of Lee et al. [36] and operationalized behavioral intention with three items drawn from the study by Kim [12] and Vilnai-Yavetz and Rafaeli [62]. All these constructs were assessed using a 7-point scale, with response options ranging from 1, indicating “Strongly disagree”, to 7, representing “Strongly agree”. In a final procedural step, in accordance with the methodology advocated by Podsakoff et al. [63], the survey items were intentionally presented in a randomized order. This methodological adjustment was implemented to minimize the potential influence of common method variance on the study’s findings. 

## 4. Results

### 4.1. Test of Reliability and Validity

As evidenced by the seminal work of Anderson and Gerbing [64], our investigative endeavor adheres to a methodologically rigorous two-step approach for the purpose of data analysis preceding the implementation of structural equation modeling (SEM). Initially, this study undertook an evaluation and affirmation of reliability and validity using two pivotal statistical techniques, as outlined in the works of Anderson and Gerbing [64] and Hair et al. [65]. The first of these techniques involved the computation of Cronbach’s alpha coefficients for each variable utilizing the software SPSS 28.0, while the second encompassed the application of confirmatory factor analysis (CFA) employing AMOS 28.0 to scrutinize the proposed measurement model. This process was conducted prior to embarking on the exploration of substantive relationships among the variables central to this research.

In accordance with the findings presented in Table 2, during the initial phase, all variables displayed Cronbach’s alpha coefficients surpassing the established threshold of 0.70, a widely accepted benchmark for establishing reliability within the domain of the social sciences [12,65]. Subsequently, in the second phase, our research conducted a comprehensive CFA to evaluate the conceptual performance of the measurement model, encompassing all corresponding indicators, following an exhaustive assessment of validity. During this phase, six items with negative impacts on validity were deleted based on the modification indices generated by AMOS 28.0. The fit indices of the measurement model, as ascertained, were as follows: χ^2^ = 2962.907, degrees of freedom = 1146, *p* < 0.001, Root Mean Square Error of Approximation (RMSEA) = 0.054, Normed Fit Index (NFI) = 0.868, Comparative Fit Index (CFI) = 0.914, and Tucker-Lewis Index (TLI) = 0.905. Collectively, these fit indices unequivocally demonstrated a robust conceptual alignment with the measurement model. Furthermore, it is noteworthy that all indicators exhibited standardized factor loadings exceeding 0.45 at a statistically significant level of *p* < 0.01, thus corroborating the convergent validity of each construct, as documented in Table 2 [65]. Detailed information regarding the assessment of composite construct reliability is presented in Table 2, taking into account the outcomes of the aforementioned CFA procedure.

In the present research, a thorough examination was conducted, entailing a comprehensive correlation analysis that considered all variables within the study. The principal objective of this analytical endeavor was to discern the presence of discriminant validity by ascertaining the proportions of the average variance extracted for each variable. It is imperative to underscore that even in cases where the calculated average variance extracted value falls below the conventional threshold of 0.50, it may still manifest discriminant validity if it surpasses the corresponding squared correlations among variables, as articulated in the work of Kim and Stepchenkova [61]. Consequently, as illustrated in Table 3, it is of significance to observe that the average variance extracted values pertaining to all variables notably exceeded the squared correlations associated with other variables, thereby firmly establishing their discriminant validity.

Before delving into the examination of our hypothesized relationships, our research judiciously employed Harman’s one-factor test as a preemptive safeguard to assess the efficacy of mitigating common method bias in accordance with the recommendations put forth by Podsakoff et al. [63]. The meticulous implementation of procedural and statistical safeguards serves as an integral component of our research design, assuring the prevention of potential misinterpretations stemming from the confounding influence of common method bias. Harman’s one-factor test operates on the premise that a multi-dimensional model (or measurement model) should yield more favorable χ^2^ and degrees of freedom values in comparison to those obtained from a one-factor model. Essentially, when a one-factor model outperforms a multi-dimensional model in explicating all observed indicators, it raises substantive concerns regarding the presence of common method bias. Within the scope of our study, our measurement model yielded χ^2^ = 2962.907 with degrees of freedom = 1146. In stark contrast, the one-factor model produced χ^2^ = 7394.850 with degrees of freedom = 1224. This notable statistical disparity unequivocally underscores the effectiveness of the procedural measure we instituted in controlling for common method bias within the confines of this research endeavor.

### 4.2. Test of Research Hypotheses

Following the two-step for evaluation of the measurement model’s reliability and validity, our research performed SEM via the covariance matrix and maximum likelihood estimates with AMOS 28.0 to empirically explore the hypothesized paths. The measurement model’s goodness-of-fit indices were meticulously scrutinized, generating the following results: χ^2^ = 2999.908, degrees of freedom = 1164, *p* < 0.001, RMSEA = 0.054, NFI = 0.867, CFI = 0.914, and TLI = 0.905 in accordance with Hair et al. [65].

The empirical findings of SEM showcased that negative emotion was significantly influenced by several sub-dimensions of e-servicescapes, such as security (standardized estimate = −0.191, critical ratio = −3.036, *p* < 0.01), visual appeal (standardized estimate = −0.211, critical ratio = −2.090, *p* < 0.05), entertainment value (standardized estimate = −0.508, critical ratio = −3.785, *p* < 0.01), originality of design (standardized estimate = −0.220, critical ratio = −1.657, *p* < 0.10), and social factors (standardized estimate = −0.434, critical ratio = −3.934, *p* < 0.01), except for usability, customization, interactivity, and relevance of information, lending support to H1-3, H1-5, H1-9, H1-13, H1-17 only. Moreover, the empirical results of SEM addressed that positive emotion was significantly affected by visual appeal (standardized estimate = 0.328, critical ratio = 1.676, *p* < 0.10), customization (standardized estimate = 0.261, critical ratio = 2.056, *p* < 0.05), interactivity (standardized estimate = 0.365, critical ratio = 2.148, *p* < 0.05), and relevance of information (standardized estimate = 0.320, critical ratio = 2.075, *p* < 0.05), except for usability, security, entertainment value, originality of design, and social factors, supporting H1-6, H1-8, H1-12, H1-16 only. Furthermore, the empirical outcomes demonstrated that satisfaction was markedly influenced by both positive (standardized estimate = 0.355, critical ratio = 6.101, *p* < 0.01) and negative emotions (standardized estimate = −0.406, critical ratio = −7.239, *p* < 0.01), leading supports to H2-1 and H2-2. Lastly, behavioral intention was significantly shaped by positive emotion (standardized estimate = 0.174, critical ratio = 3.048, *p* < 0.01), negative emotion (standardized estimate = −0.425, critical ratio = −7.331, *p* < 0.01), and satisfaction (standardized estimate = 0.371, critical ratio = 6.217, *p* < 0.01), supporting H3-1, H3-2, and H4 (see Table 4).

The finding that security significantly influences negative emotions underscores the critical role of perceived safety and data protection in shaping users’ emotional responses. Security concerns can lead to feelings of anxiety and discomfort, detracting from the overall user experience. Prior research has indicated that perceived security is a pivotal determinant of user trust and engagement in digital environments [17,18]. Enhancing security measures, such as robust encryption and transparent data handling practices, and clearly communicating these measures to users can mitigate these negative emotions and foster a sense of trust and safety.

Visual appeal’s significant impact on negative emotions highlights the importance of aesthetic design in digital interfaces. A visually unappealing interface can cause frustration and dissatisfaction, as users might find it difficult to navigate or unpleasant to use. This aligns with the environmental psychology theory, which posits that physical and visual stimuli in an environment influence emotional states and behaviors [12,15]. Therefore, investing in aesthetically pleasing designs that enhance visual clarity and attractiveness is essential for reducing negative emotional responses.

Entertainment value has a substantial negative effect on negative emotions, indicating that when users find the interaction entertaining, their negative emotions are significantly reduced. This finding supports the notion that engaging and enjoyable content can distract from potential frustrations or concerns, thereby improving the overall user experience [12,19]. Developers should focus on incorporating elements that enhance the entertainment value of ChatGPT, such as gamified features, interactive content, and humor.

The originality of design significantly affects negative emotions, suggesting that innovative and creative design features can alleviate negative feelings. This aligns with the concept that novelty and uniqueness in design can capture users’ interest and enhance their emotional engagement [12,15]. By continuously innovating and incorporating unique design elements, developers can keep the user experience fresh and engaging, reducing negative emotional responses.

Social factors significantly influence negative emotions, indicating that the perceived human-like interaction and social presence of ChatGPT can mitigate negative feelings. This finding is consistent with research suggesting that social interactions and the perception of empathy and warmth from AI can enhance user experience and reduce feelings of isolation or frustration [12,15]. Enhancing the social aspects of ChatGPT, such as incorporating more personalized and empathetic responses, can significantly improve user satisfaction.

Visual appeal also significantly impacts positive emotions, emphasizing that an aesthetically pleasing interface not only reduces negative emotions but also enhances positive ones. This dual impact underscores the critical role of visual design in user experience. Research indicates that visual aesthetics contribute to user satisfaction by providing a pleasant and engaging interaction [12,15]. Therefore, maintaining high visual standards and continually refining the aesthetic elements of ChatGPT can foster positive emotional responses and increase user engagement.

Customization’s significant effect on positive emotions highlights the value users place on personalized experiences. The ability to tailor interactions to individual preferences enhances user satisfaction by making the experience more relevant and engaging [18,24]. Personalization can be achieved through features such as adaptive learning algorithms that adjust responses based on user behavior and preferences, further enhancing the user experience.

Interactivity significantly influences positive emotions, suggesting that dynamic and engaging interactions are crucial for fostering positive user experiences. Interactive elements, such as responsive feedback, real-time engagement, and interactive prompts, can make the user feel more involved and invested in the interaction [17,24]. Enhancing the interactivity of ChatGPT can lead to greater user satisfaction and continued usage.

The relevance of information significantly impacts positive emotions, indicating that users highly value accurate and pertinent information. Providing relevant and useful information enhances user trust and satisfaction, making the interaction more valuable [17,19]. Ensuring that ChatGPT delivers high-quality, relevant content tailored to user needs is essential for fostering positive emotional responses.

The insights gained from these findings have important implications for the future development of ChatGPT. Specifically, they suggest that developers should (1) *Enhance Security*: Prioritize robust security measures and transparent communication to build user trust and reduce negative emotions, (2) *Invest in Visual Design*: Maintain high visual standards to enhance both positive and negative emotional responses, (3) *Increase Customization and Interactivity*: Provide personalized and engaging interactions to foster positive emotions and satisfaction, (4) *Ensure Relevance of Information*: Deliver accurate and pertinent information to enhance user trust and satisfaction, and (5) *Foster Social and Entertainment Value*: Incorporate features that enhance social presence and entertainment to improve overall user experience.

Moreover, our research investigated indirect pathways from the sub-dimensions of e-servicescapes to behavioral intention using the Monte Carlo and Bias Corrected bootstrapping techniques. According to Tofighi and Kelley [66], these statistical methodologies estimate the significance levels and confidence intervals for each indirect influence by thoroughly calculating the direct, indirect, and total effects among independent variables, multiple mediators, and dependent variables. As presented in Table 4, among the sub-dimensions of e-servicescapes, security (*p* < 0.10), visual appeal (*p* < 0.05), customization (*p* < 0.10), interactivity (*p* < 0.10), and originality of design (*p* < 0.05), exhibited statistically significant indirect influences on behavioral intention.

## 5. Conclusions and Implications

### 5.1. Theoretical Implications

Our empirical research contributes significantly to the theoretical foundations of environmental psychology and servicescapes theory. This contribution extends the boundaries of these theories from their traditional focus on physical and online environments to encompass the context of virtual interactions, particularly within the domain of AI-powered chatbots, such as ChatGPT. Specifically, the concept of servicescapes, originally grounded in environmental psychology theory, has predominantly been examined in physical settings or within general online contexts, such as websites [12,13,14,15]. However, our current study underscores the adaptability of this concept to the realm of virtual interactions with AI chatbots like ChatGPT. In this context, it becomes evident that negative and positive emotional responses are distinctly influenced by a unique set of virtual elements and cues. These elements, both conceptually and empirically identified, encompass aspects such as usability, security, visual appeal, customization, entertainment value, interactivity, originality of design, relevance of information, and social factors [12,15,17,18,19,24]. This innovative application and expansion of servicescapes theory demonstrate its enduring relevance within the realm of environmental psychology and its capacity to elucidate how chatbot users perceive and respond to their digital interactions with AI-powered services and products [2]. Thus, our research enriches the theoretical landscape of this field.

Moreover, our study not only substantiates the presence of users’ emotional responses within the sphere of virtual interactions with AI-powered chatbots but also underscores the critical differentiation between negative and positive emotions [20,21]. Through the identification of specific sub-dimensions within the e-servicescape that evoke these discrete emotional responses among ChatGPT users, our research contributes to a refined theoretical comprehension of the intricate interplay between users’ emotional states and the constituent elements of interaction-based virtual environments [12,20]. This conceptual and empirical differentiation facilitates an in-depth exploration by scholars in the chatbot domain into the nuanced emotional dynamics that occur during digital interactions with AI-powered technologies, thereby laying the foundation for more precise theoretical models and psychological frameworks [20,21]. Furthermore, our study firmly establishes the indispensable roles of users’ emotions in shaping their favorable behavioral inclinations, particularly in the context of AI-powered technology such as ChatGPT [21,48]. The empirical findings of our research underscore that positive emotions, such as pleasure and joy, wield a substantial influence on ChatGPT users’ intentions to sustain their engagement with this technology. Conversely, negative emotions, such as disappointment and frustration, exert a deterrent effect on users’ proclivity for further interaction with AI chatbots [48]. Consequently, our work advances our theoretical comprehension of the psychological nexus between users’ emotions and user-centric design, underscoring the imperative to incorporate emotional considerations in the developmental and implementation processes of AI-powered chatbots or virtual agents.

### 5.2. Managerial Implications

The empirical findings derived from our research offer valuable practical insights applicable to developers and businesses aiming to enhance the functionality and design of AI-powered chatbots, such as ChatGPT. Specifically, our results underscore the significance of prioritizing certain dimensions, including visual appeal, usability, interactivity, and the relevance of information, to cultivate more favorable e-servicescapes. By doing so, developers can effectively elevate users’ emotional responses and overall experiences. Practitioners can readily apply these insights to fine-tune their chatbot interfaces, thereby creating visually captivating and user-friendly environments that foster user engagement. Additionally, our study emphasizes the pivotal role of user-centric design principles in the development of AI-powered chatbots. Developers should direct their efforts towards crafting aesthetically pleasing, user-friendly, and interactive interfaces, as these dimensions of e-servicescapes exert a direct influence on users’ emotional responses, satisfaction levels, and subsequent intentions to sustain engagement with such technology. Implementation of user-centered design principles at the managerial level holds the potential to enhance long-term user retention and engagement. Furthermore, organizations should prioritize ensuring the security and reliability of information dispensed by AI-powered chatbots to maintain user trust and credibility. Our research indicates that ChatGPT users place trust in and continue using such technology when they perceive information security and response accuracy. Practical implications encompass the integration of robust data security measures, transparency in data handling, and continuous refinement of AI-powered chatbots’ accuracy through advanced machine learning algorithms.

From a practical marketing perspective, our research insights can guide the development and implementation of strategies aimed at eliciting positive emotional responses during interactions with AI-powered chatbots like ChatGPT. Strategies may encompass the creation of engaging and personalized content that highlights the value users derive from the technology, resulting in elevated positive emotions, heightened user satisfaction, and increased loyalty. Marketing campaigns can thus emphasize the benefits and personalized functionalities of AI-powered chatbots. Furthermore, it is imperative for marketers and developers to institute mechanisms for ongoing monitoring of user emotions and satisfaction with the technology. Consequently, the establishment of feedback channels becomes crucial to gather user input and identify areas necessitating improvement for both current and potential users. Addressing issues that engender negative emotions can effectively mitigate user churn and foster retention. Practically, marketers and developers must actively participate in the setup of feedback mechanisms, frequent analysis of user feedback, and iterative enhancements to the technology. 

Lastly, enhancing AI literacy among users is essential for maximizing the benefits of AI-powered chatbots. Social media channels, such as YouTube, can serve as valuable resources for individuals seeking to improve their understanding and utilization of AI technologies. For instance, developers can create a series of instructional videos on YouTube that cover topics such as how to customize ChatGPT settings, use advanced features, and troubleshoot common issues. Additionally, offering live Q&A sessions and webinars on platforms like YouTube can provide real-time assistance and foster a community of informed users. Providing accessible tutorials, guides, and educational content on platforms like YouTube can empower users with the necessary skills to navigate and optimize their interactions with ChatGPT. This educational outreach can reduce negative emotions associated with the perceived complexity of AI and ultimately enhance overall user satisfaction and behavioral intentions. Examples of content could include step-by-step guides on setting up and using ChatGPT, explanations of how AI algorithms work in simple terms, and user testimonials sharing successful use cases.

### 5.3. Limitations and Directions for Future Research

One notable limitation of our research pertains to the generalizability of empirical findings, given its focus on a specific user context, namely ChatGPT. Consequently, the results derived from this study may not be universally applicable to all AI-powered chatbots. In addition, the results derived from this study may not be universally applicable to all features of ChatGPT, such as voice, image, or video capabilities. Specifically, our study primarily investigated the text-based feature of ChatGPT, and therefore, the findings cannot be generalized to other modalities of the AI system. Future research should consider exploring the effects of ChatGPT’s various features to provide a more comprehensive understanding of user interactions across different modalities. Therefore, it is imperative for future research endeavors to explore diverse chatbot platforms and features encompassing various user demographics, cultural backgrounds, or specific industry contexts to ascertain the broader implications of our findings. Another noteworthy limitation concerns the reliance on self-reported data as the primary means of assessing ChatGPT users’ emotional states and satisfaction levels. Self-reporting methods inherently carry the risk of introducing response bias and may not consistently provide an accurate reflection of user experiences. As a result, forthcoming studies should incorporate objective measures within a mixed-method approach, combining self-reported data with the collection of actual behavioral data or physiological datasets to offer a more comprehensive and balanced assessment. Lastly, the cross-sectional design employed in this study has its inherent limitations in capturing temporal changes. Longitudinal research, on the other hand, offers the potential to provide insights into the evolution of user emotions and satisfaction over extended interactions with AI-powered chatbots. Such research can shed light on whether these changes exert varying influences on user behavior over time. By systematically addressing these aforementioned limitations and considering the proposed research directions, the academic field can significantly advance its understanding of e-servicescapes, user emotions, and behavior in the context of AI-powered chatbot interactions. Ultimately, this progress will contribute to the more effective design and deployment of such technology in virtual environments.

## Figures and Tables

**Table 1 behavsci-14-00558-t001:** Demographic analysis of respondents.

Demographic Variables		Frequency (Percent)
Gender	Female	275 (50.2%)
	Male	273 (49.8%)
Age	18–29	391 (71.4%)
	30–39	137 (25.0%)
	40–49	12 (2.2%)
	50 or above	8 (1.5%)
Education	High school graduate	188 (34.3%)
	Working on or completed associate degree	162 (29.6%)
	Working on or completed bachelor’s degree	161 (29.4%)
	Working on or completed graduate degree	37 (6.7%)
Frequency of ChatGPT usage per week	1–5 times	90 (16.4%)
	5–10 times	94 (17.2%)
	11–20 times	136 (24.8%)
	Over 10 times	228 (41.6%)
Occupation	Self-employed	125 (22.8%)
	Employed	392 (71.5%)
	Student	31 (5.7%)

**Table 2 behavsci-14-00558-t002:** Measurement model from CFA.

Constructs and Items	Standardized Estimates	Critical Ratios
** *Usability (α = 0.924) from Kim [12] and Tankovic and Benazic [19]* **		
The functions within ChatGPT are straightforward to navigate.	0.771	Fixed
ChatGPT is designed with user-friendliness in mind.	0.687	16.831
In general, I find ChatGPT easily usable for my tasks.	0.796	20.092
ChatGPT clearly presents links and destinations.	0.828	21.090
ChatGPT provides convenient methods for moving between related functions.	0.843	21.579
Navigating through ChatGPT feels intuitively logical.	0.864	22.239
ChatGPT includes navigation aids.	0.791	19.924
** *Security (α = 0.921) from Teng et al. [17] and Tran and Strutton [18]* **		
ChatGPT incorporates effective security measures.	0.762	Fixed
The security protocols of ChatGPT appear robust.	0.894	22.955
I harbor no security apprehensions when using ChatGPT.	0.897	22.969
In general, ChatGPT demonstrates a strong commitment to security.	0.917	23.571
** *Visual appeal (α = 0.804) from Kim [12] and Yadav and Mahara [15]* **		
ChatGPT exhibits a visually pleasing design.	0.645	Fixed
I find the appearance of ChatGPT appealing.	0.740	14.652
ChatGPT possesses an attractive visual layout.	0.748	14.780
The manner in which ChatGPT presents its features is visually appealing	0.770	15.123
** *Customization (α = 0.907) from Tran and Strutton [18] and Wu et al. [24]* **		
The services provided by ChatGPT are frequently tailored to my preferences.	0.822	Fixed
I perceive ChatGPT as being crafted with my needs in mind.	0.857	24.037
ChatGPT treats me as an individual user.	0.889	25.408
I have the option to customize ChatGPT according to my preferences if I desire.	0.805	21.920
** *Entertainment value (α = 0.833) from Kim [12] and Tankovic and Benazic [19]* **		
I engage with ChatGPT primarily for my own enjoyment.	0.665	Fixed
I find ChatGPT highly entertaining.	0.605	12.846
I take pleasure in using ChatGPT for its intrinsic value, not solely because I acquired it.	0.824	16.760
ChatGPT not only aids in my tasks but also provides entertainment.	0.645	13.602
ChatGPT’s enthusiasm is infectious and uplifts my experience.	0.816	16.636
** *Interactivity (α = 0.759) from Teng et al. [17] and Wu et al. [24]* **		
I perceive ChatGPT as dynamic.	0.547	Fixed
I experience a high level of engagement with ChatGPT.	0.780	12.444
ChatGPT provides diverse perspectives on information.	0.846	12.885
ChatGPT includes effective search tools to help me locate and accomplish my desires. *	-	-
** *Origin of design (α = 0.752) from Kim [12] and Yadav and Mahara [15]* **		
ChatGPT is characterized by freshness and originality.	0.620	Fixed
ChatGPT demonstrates innovation and creativity.	0.794	14.183
Engaging with ChatGPT feels adventurous.	0.736	13.514
ChatGPT is advanced in its design and features. *	-	-
** *Relevance of information (α = 0.740) from Tankovic and Benazic [19] and Teng et al. [17]* **		
Each feature of ChatGPT clearly communicates what one can anticipate or accomplish.	0.756	Fixed
Visual information and data regarding topics are readily accessible with ChatGPT.	0.653	15.475
All pertinent information is readily accessible through ChatGPT. *	-	-
There is an abundance of pertinent information available through ChatGPT.	0.688	16.386
Technical details about ChatGPT can be easily accessed. *	-	-
** *Social factors (α = 0.888) from Kim [12] and Yadav and Mahara [15]* **		
I sense a human-like touch when I interact with ChatGPT.	0.843	Fixed
There is a potential for connecting with other users through ChatGPT.	0.774	21.260
Interacting with ChatGPT gives me a sense of friendliness.	0.813	22.902
I feel a sense of belonging when I interact with ChatGPT.	0.831	23.726
ChatGPT exhibits a human-like warmth. *	-	-
I perceive a human-like sensitivity in ChatGPT. *	-	-
** *Negative emotion (α = 0.877)* **		
While interacting with ChatGPT, I feel…		
Bored	0.692	Fixed
Angry	0.849	18.153
Sleepy	0.824	17.671
Annoyed	0.866	18.451
** *Positive emotion (α = 0.939)* **		
While interacting with ChatGPT, I feel…		
Happy	0.891	Fixed
Energetic	0.905	32.257
Excited	0.893	31.263
Relaxed	0.876	29.911
** *Satisfaction (α = 0.809)* **		
I feel very good with ChatGPT while interacting with it.	0.787	Fixed
I am content with my decision to use ChatGPT for my information and conversation needs.	0.859	19.612
Overall, I am satisfied with my interactions with ChatGPT.	0.683	15.842
** *Behavioral intention (α = 0.761)* **		
I would like ChatGPT to assist me with a wide range of tasks.	0.776	Fixed
I find interacting with ChatGPT enjoyable.	0.471	10.301
I prefer that ChatGPT continues to assist me with various tasks in the future.	0.721	16.014

Note: An asterisk (*) denotes a deleted item.

**Table 3 behavsci-14-00558-t003:** Construct intercorrelations (*Φ*), average variance extracted, and composite construct reliability.

Construct	1	2	3	4	5	6	7	8	9	10	11	12	13
1. Usability	1												
2. Security	0.639 **	1											
3. Visual appeal	0.655 **	0.691 **	1										
4. Customization	0.629 **	0.686 **	0.653 **	1									
5. Entertainment value	0.655 **	0.672 **	0.653 **	0.679 **	1								
6. Interactivity	0.586 **	0.529 **	0.571 **	0.396 **	0.573 **	1							
7. Originality of design	0.573 **	0.445 **	0.544 **	0.299 **	0.457 **	0.637 **	1						
8. Relevance of information	0.642 **	0.645 **	0.666 **	0.550 **	0.668 **	0.584 **	0.593 **	1					
9. Social factors	0.628 **	0.705 **	0.646 **	0.627 **	0.681 **	0.575 **	0.515 **	0.693 **	1				
10. Negative emotion	−0.611 **	−0.730 **	−0.593 **	−0.696 **	−0.637 **	−0.480 **	−0.338 **	−0.624 **	−0.683 **	1			
11. Positive emotion	0.632 **	0.663 **	0.610 **	0.577 **	0.617 **	0.574 **	0.533 **	0.676 **	0.659 **	−0.602 **	1		
12. Satisfaction	0.458 **	0.474 **	0.425 **	0.404 **	0.513 **	0.458 **	0.361 **	0.504 **	0.549 **	−0.546 **	0.546 **	1	
13. Behavioral intention	0.475 **	0.542 **	0.446 **	0.465 **	0.529 **	0.472 **	0.327 **	0.476 **	0.550 **	−0.550 **	0.550 **	0.510 **	1
Composite construct reliability	0.925	0.874	0.787	0.876	0.839	0.774	0.762	0.742	0.906	0.884	0.939	0.822	0.701
Average variance extracted	0.638	0.642	0.505	0.652	0.514	0.541	0.519	0.490	0.708	0.657	0.794	0.608	0.448

** *p* < 0.01, * *p* < 0.05.

**Table 4 behavsci-14-00558-t004:** Standardized structural estimates.

Paths		Standardized Estimates	Standardized Errors	Critical Ratios
H1-1	Usability → Negative emotion	−0.099	0.088	−1.249
H1-2	Usability → Positive emotion	0.123	0.152	0.854
H1-3	Security → Negative emotion	−0.191	0.077	−3.036 ***
H1-4	Security → Positive emotion	0.123	0.135	1.065
H1-5	Visual appeal → Negative emotion	−0.211	0.116	−2.090 **
H1-6	Visual appeal → Positive emotion	0.328	0.213	1.676 *
H1-7	Customization → Negative emotion	−0.074	0.073	−1.075
H1-8	Customization → Positive emotion	0.261	0.128	2.056 **
H1-9	Entertainment value → Negative emotion	−0.508	0.156	−3.785 ***
H1-10	Entertainment value → Positive emotion	0.369	0.326	1.258
H1-11	Interactivity → Negative emotion	−0.045	0.110	−0.519
H1-12	Interactivity → Positive emotion	0.365	0.205	2.148 **
H1-13	Originality of design → Negative emotion	−0.220	0.139	−1.657 *
H1-14	Originality of design → Positive emotion	0.321	0.286	1.118
H1-15	Relevance of information → Negative emotion	−0.014	0.306	−0.054
H1-16	Relevance of information → Positive emotion	0.320	0.272	2.075 **
H1-17	Social factors → Negative emotion	−0.434	0.112	−3.934 ***
H1-18	Social factors → Positive emotion	0.304	0.238	1.240
H2-1	Positive emotion → Satisfaction	0.355	0.059	6.101 ***
H2-2	Negative emotion → Satisfaction	−0.406	0.054	−7.239 ***
H3-1	Positive emotion → Behavioral intention	0.174	0.051	3.048 ***
H3-2	Negative emotion → Behavioral intention	−0.425	0.050	−7.331 ***
H4	Satisfaction → Behavioral intention	0.371	0.053	6.217 ***
**Indirect paths**		**Unstandardized** **estimates**	**95% bootstrapping confidence intervals**	***p*-values**
	Usability → Behavioral intention	0.090	−0.049~0.293	0.203
	Security → Behavioral intention	0.156	−0.033~0.275	0.078
	Visual appeal → Behavioral intention	0.224	−0.499~−0.048	0.023
	Customization → Behavioral intention	0.126	−0.011~0.306	0.052
	Entertainment value → Behavioral intention	0.085	−0.346~0.433	0.249
	Interactivity → Behavioral intention	0.181	−0.025~0.449	0.090
	Originality of design → Behavioral intention	0.202	−0.634~−0.001	0.050
	Relevance of information → Behavioral intention	0.409	−0.101~0.528	0.123
	Social factors → Behavioral intention	0.137	−0.236~0.327	0.246

*** *p* < 0.01, ** *p* < 0.05, * *p* < 0.10.

## Data Availability

The data presented in this study are available on request from the corresponding author. The data are not publicly available due to privacy reasons.

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
