# Peer review of "Unveiling the e-Servicescape of ChatGPT: Exploring User Psychology and Engagement in AI-Powered Chatbot Experiences"

_behavsci, 2024, doi:10.3390/bs14070558_

Round 1
Reviewer 1 Report
Comments and Suggestions for Authors
This paper addresses an important and emerging topic in ChatGPT and its psychological impacts on users, both positive and negative. It employs a mixed-methodology study using an online survey with a substantial sample of 548 participants from the United States. The survey aimed to test four hypotheses regarding the effects of e-servicescape on ChatGPT users' emotions. The results indicated that various e-servicescape sub-dimensions influenced negative emotions, while factors such as visual appeal and customization influenced positive emotions. Both types of emotions significantly impacted user satisfaction and their intention to continue engaging with ChatGPT.
While there are successful elements in this manuscript that demonstrate good mixed-methodology research including motivation and research settings, several aspects require improvement:
- The study should be more focused and synthesized through a more targeted mixed-methodology approach. The results should systematically address the hypotheses and explain the significance of the statistically significant findings for e-servicescape and future developments in ChatGPT.
- The manuscript needs a more comprehensive literature review. There is a significant body of literature on AI chatbots, their development, and their impact on human behavior, including decision-making, privacy, and emotions. This includes generative AI and social chatbots like Replika AI. Several citations are misinterpreted, such as Rajaobelina et al. (2021), which is incorrectly cited regarding negative emotions in chatbots. The author should accurately cite this work, highlighting aspects such as privacy concerns, emotion building (e.g., trust, loyalty), and technology anxiety. Additionally, Rajaobelina et al. generalize about rule-based personal assistant chatbots, not generative AI chatbots like ChatGPT.
- The study’s data collection methods have limitations. While MTurk is useful for finding participants, literature shows its participants may not be highly representative. The reliability testing is accurate; however, the author needs to clarify why this study can be considered representative of the United States. Details about any compensation given to participants should be included. Additionally, ethical considerations, including the consent mechanism, should be highlighted.
- In the limitations section (5.3), it should be emphasized that the results cannot be generalized to all features of ChatGPT (e.g., voice, image, video) but only cover its text-based feature. If the manuscript aims to cover all features of ChatGPT, this needs to be addressed in the literature review (Section 2) and data collection (Section 3.1). Furthermore, the manuscript should clarify which version of ChatGPT (e.g., GPT-3, GPT-3.5, GPT-4, GPT-4.0) is being studied.
- The theoretical and future research implications for marketers and developers are well interpreted. However, Section 5.2, which discusses AI literacy, should emphasize that social media channels such as YouTube can be valuable resources for individuals seeking to enhance their AI literacy.
Overall, that is a valuable study for the emerging ChatGPT studies. While this manuscript has potential, addressing these points will strengthen the work significantly and make it more suitable for publication. This manuscript needs specifically more work on literature review, interpreting the results, and conclusion sections need more work since the descriptive data has more things to be said for this study.
Author Response
Please see the attached response letter.

Reviewer 2 Report
Comments and Suggestions for Authors
This study defines the concept of "e-servicescape" in the context of chatGPT and examines its relationship with emotions, satisfaction, and behavioral intention. The following parts need to be modified.
1. This study uses the terms "e-servicescape" and "virtual servicescape" interchangeably. To ensure clarity and consistency, it is necessary to standardize the usage of these terms throughout the study.
2. In the measurement section, insights were sought from three scholars specializing in digital marketing. It is important to briefly outline their backgrounds and relevant research experiences.
3. It is essential to specify the 5 deleted items from Table 2.
4. Detailed explanations are required for the 9 constructs listed in Table 2 (e.g., factor, security, visual appeal...). Additionally, specify the previous studies from which these constructs and their corresponding items were derived.
Author Response

(The authors gave the same response as above.)

Round 2
Reviewer 1 Report
Comments and Suggestions for Authors
Thank you for addressing the concerns raised earlier in the review. The author improved the previous manuscript very well. This version with a revised title is a good work for this area. The minor issue is that the URL should be in the footnote "According to Namepepper in 2024 (https://www.namepepper.com/chatgpt-users)". Hope to see this manuscript in the journal.
Author Response
Please find the attached response letter.
